# Prevalence of neutralising antibodies against SARS-CoV-2 in acute infection and convalescence: A systematic review and meta-analysis

**Helen R. Savage**[1], **Victor S. Santos**[2,3,4], **Thomas Edwards**[1], **Emanuele Giorgi**[5], **Sanjeev Krishna**[6,7,8], **Timothy D. Planche**[6,9], **Henry M. Staines**[6], **Joseph R. A. Fitchett**[10,11], **Daniela E. Kirwan**[6], **Ana I. Cubas Atienzar**[1], **David J. Clark**[6], **Emily R. Adams**[1], **Luis E. Cuevas**[1,12]*

1 Departments of Clinical Sciences and Tropical Disease Biology, Liverpool School of Tropical Medicine, Liverpool, United Kingdom, 2 Núcleo de Epidemiologia e Saúde Pública, Universidade Federal de Alagoas, Arapiraca, Brazil, 3 Programa de Pós-Graduação em Ciências da Saúde, Universidade Federal de Alagoas, Maceió, Brazil, 4 Programa de Pós-Graduação em Ciências da Saúde, Universidade Federal de Sergipe, Aracaju, Brazil, 5 Lancaster Medical School, Lancaster, United Kingdom, 6 St. George's, University of London, London, United Kingdom, 7 Universitätsklinikum Tübingen, Tübingen, Germany, 8 Centre de Recherches Médicales de Lambaréné, Lambaréné, Gabon, 9 St. George's University Hospitals National Health Services Foundation Trust, London, 10 Mologic, Thurleigh, United Kingdom, 11 Harvard T.H. Chan School of Public Health, Boston, Massachusetts, United States of America, 12 Bingham University, Nasarawa State, Nigeria

* Luis.Cuevas@lstmed.ac.uk

**Data Availability Statement:** All relevant data are within the manuscript and its Supporting Information files.

## Abstract

### Background

Individuals infected with SARS-CoV-2 develop neutralising antibodies. We investigated the proportion of individuals with SARS-CoV-2 neutralising antibodies after infection and how this proportion varies with selected covariates.

### Methodology/Principal findings

This systematic review and meta-analysis examined the proportion of individuals with SARS-CoV-2 neutralising antibodies after infection and how these proportions vary with selected covariates. Three models using the maximum likelihood method assessed these proportions by study group, covariates and individually extracted data (protocol CRD42020208913). A total of 983 reports were identified and 27 were included. The pooled (95%CI) proportion of individuals with neutralising antibodies was 85.3% (83.5–86.9) using the titre cut off >1:20 and 83.9% (82.2–85.6), 70.2% (68.1–72.5) and 54.2% (52.0–56.5) with titres >1:40, >1:80 and >1:160, respectively. These proportions were higher among patients with severe COVID-19 (e.g., titres >1:80, 84.8% [80.0–89.2], >1:160, 74.4% [67.5–79.7]) than those with mild presentation (56.7% [49.9–62.9] and 44.1% [37.3–50.6], respectively) and lowest among asymptomatic infections (28.6% [17.9–39.2] and 10.0% [3.7–20.1], respectively). IgG and neutralising antibody levels correlated poorly.

**Funding:** This study received funding from the UK Research Council through a PhD scholarship from the MRC Doctoral Training Partnership to HRS. LEC and ERA were funded by the UK National Institute for Health Research Health Protection Research Unit in Emerging and Zoonotic Infections, the Centre of Excellence in Infectious Diseases Research (Liverpool) and the Alder Hey Charity. Funding was also received from Wellcome/DFID through a grant for the development of COVID-19 diagnostics (grant number 220764/Z/20/Z to JRF) and Wellcome Trust Institutional Strategic Support Fund (204809/Z/16/Z to SK). The Rosetrees Trust and the John and Maureen Hendricks Charitable Foundation (grant number M959) was given to HMS, TP, SK, JRAF, DJC. The funders had no role in study design, data collection and analysis, decision to publish, or preparation of the manuscript.

**Competing interests:** I have read the journal's policy and the authors of this manuscript have the following competing interests: SK is advisor and together with HMS shareholders in QuantuMDx, a molecular nucleic acid test-based diagnostic company. SK is also member of the Scientific Advisory Committee for the Foundation for Innovative New Diagnostics (FIND), a not-for-profit organisation that produces global guidance on affordable diagnostics. The views expressed here are personal opinions and do not represent the recommendations of FIND. JF is an employee and shareholder of Mologic, a private biotechnology company, and a pro bono director of Global Access Diagnostics, a social enterprise delinked from commercial return. All other authors have no conflicts of interest to declare.

## Conclusions/Significance

85% of individuals with proven SARS-CoV-2 infection had detectable neutralising antibodies. This proportion varied with disease severity, study setting, time since infection and the method used to measure antibodies.

### Author summary

Severe Acute Respiratory Syndrome Coronavirus-2 (SARS-CoV-2) elicits adaptive immunological responses, including immunoglobulins A, M, and G and neutralising antibodies. Neutralising antibodies are considered markers of functional immunity and protection. However, not all individuals with proven infections have detectable neutralising antibodies. In this systematic review, we investigated the proportion of individuals with former SARS-CoV-2 infections who develop neutralising antibodies, whether their titres vary with disease severity, and their correlation with Immunoglobulin G. We found that approximately 85% of individuals with SARS-CoV-2 infection have detectable neutralising antibodies. This proportion was higher among patients with severe Coronavirus Disease 19 and lower in asymptomatic infections. The variation across studies reflected the wide range of methods used to measure both immunoglobulins and neutralising antibodies, and highlight the need for an international reference standard to measure SARS-CoV-2 antibodies.

## Introduction

The emergence of Coronavirus Disease 2019 (COVID-19), caused by the Severe Acute Respiratory Syndrome Coronavirus-2 (SARS-CoV-2), in December 2019 [1] marked the start of the first global pandemic of this century, resulting in over 34 million cases and 1 million deaths in the following six months [2]. SARS-CoV-2 infection has a wide spectrum of manifestations ranging from asymptomatic infections to a multi-system disease with multi-organ involvement and a high mortality [3]. Its diagnosis is based on the detection of viral RNA using Reverse Transcription Polymerase Chain Reaction (RT-PCR) or rapid antigen tests [4]. SARS-CoV-2 specific assays to detect immunoglobulins (Ig) G, M, and A are well established and individuals with anti-SARS-CoV-2 antibodies are expected to have some degree of protection against infection [5]. However their correlation to functional immunity is poorly described [6].

Functional immunity is better depicted by measuring neutralising antibodies, which bind to viral surface proteins, and prevent cell infection and plaque formation in cell cultures [7]. There is however a wide array of methodologies to measure neutralising antibodies, from employing microscopy to measuring cell metabolism in the presence of virus and antibody. In addition, reporting measures differ between studies, some plot a sigmoid curve and report the viral titres that reduce viral plaques or cell metabolism to 50%, and others report the minimum antibody titre that abolishes all viral activity in cell culture.

Here we present a systematic review and meta-analysis to describe the proportion of individuals who develop SARS-CoV-2 neutralising antibodies after infection, whether this proportion varies with disease severity, the time after symptoms onset, and the correlation between IgG and neutralising antibodies titres.

## Methods

This study followed the preferred reporting items for systematic reviews and meta-analyses (PRISMA) guidelines [8]. Institutional review board approval and informed consent were not required because all data were obtained from secondary data sources and were de-identified. Similarly, the study used secondary sources of data and was not possible to seek public and patient involvement. The study protocol was registered at PROSPERO (CRD42020208913).

### Search strategy and selection criteria

We conducted a systematic search of publications using the PubMed (including MEDLINE), Web of Science, and Cochrane databases, and of preprints in bioRxiv, medRxiv and the Collabovid.org website, which includes a compilation of manuscripts on COVID-19 from arXiv, bioRxiv, Elsevier and medRxiv. The search included reports from 1st January 2020 to 12th August 2020 and was limited to human studies. The search terms used were: "severe acute respiratory coronavirus 2" OR "SARS-CoV-2" OR "sars AND virus" OR "sars AND cov" OR "COVID-19" OR "COVID 2019" OR "novel coronavirus" OR "new coronavirus" OR "Wuhan coronavirus" OR "Coronavirus disease 19" OR "2019-nCoV" AND "neutralising antibod*" OR "neutralizing antibod*" OR "neutralising AND antibod*" OR "neutralizing AND antibod*", without language restrictions (S1 Table). Two reviewers (HRS and TE) independently screened the titles and abstracts and selected full text manuscripts to assess for inclusion. Studies were retained if they had tested for neutralising antibodies against SARS-CoV-2 in participants with laboratory confirmed SARS-CoV-2 infection. We included studies that reported aggregated or individual data or where data could be extracted from graphical displays. We excluded studies without original data, if data could not be extracted, or if the titre cut offs used were not comparable to other studies. Studies including SARS-CoV-2 vaccinated patients or individuals receiving plasma therapy were excluded.

### Data extraction and bias assessment

Data were extracted using a pre-piloted extraction form, including author, year, country, study design, setting (hospital, community or plasma donor), age, gender, severity of symptoms (asymptomatic, mild, moderate, severe), and weeks/months elapsed since infection. Laboratory data included the assay used to measure neutralising antibodies, the plaque reduction threshold used to classify participants as having neutralising antibodies, the lower threshold attained, additional titre thresholds used, and the IgG Enzyme linked immunoassay (ELISA) used. Data were extracted from digitised graphs using Engauge software for individual data extraction (http://markummitchell.github.io/engauge-digitizer/). Data were selected using a 50% plaque reduction cut off for neutralising antibody titres (i.e., 50% of the virus in the culture was neutralised by the patient's serum (PRNT50)) or the PRNT90 for one study in which the PRNT50 was not provided. Immunoglobulins usually increase on week after infection and peak after 21 days. Thus, for studies that described a time window for sample collection (i.e., 7–10 days) we recorded the end of the window to generate the most conservative estimate. We extracted data for titre cut offs 1:20, 1:40, 1:80, and 1:160, as these were the dilutions used by most studies.

The NIH Study Quality Assessment tool was used to assess the risk of bias and study quality for case series and cohort studies (https://www.nhlbi.nih.gov/health-topics/study-quality-assessment-tools). Studies were initially rated as having good, fair, or poor quality, and ratings were discussed to reach consensus.

### Data analysis

We developed three statistical models in an R software environment to assess 1) the combined proportion of individuals with neutralising antibodies across studies; 2) the correlation between the proportions of participants with IgG and neutralising antibodies and 3) the correlation between individual-level IgG and neutralising antibodies. The meta-analysis combined the proportion of participants with neutralising antibodies across all studies. We used the maximum likelihood method with a quasi-Monte Carlo approximation. Ten thousand samples were simulated using the inverse transformation method by conditioning on different variables that could modify the proportion positive, including disease severity, participant recruitment setting, and neutralising antibody detection method. The resulting proportions were summarised as means and 95% confidence intervals. The correlation between the proportions of participants in a study with IgG and neutralising antibodies was estimated using a quasi-Monte Carlo method for maximum likelihood estimation, adjusting for study setting, titre cut off threshold, and neutralising antibody detection method. The 95% confidence intervals were obtained via parametric bootstraps. The individual correlation model was a bivariate model of individually extracted IgG and neutralising antibody data. Each study was modelled separately because studies used different ELISA and reporting units and did not provide enough background to standardise results. The model was fitted using a maximum likelihood method, with covariates taken from each study. A more detailed description of the statistical methods is available in the supporting information (S1 Text).

## Results

The search identified 983 reports. After screening titles and abstracts 78 full-text articles were assessed for eligibility, resulting in 27 studies that met the criteria for inclusion in the analysis (Fig 1). Twelve studies were case series and fifteen cohort studies, as shown in Table 1. Seventeen studies were conducted in hospital settings, three in community or outreach centres, and seven were based on plasma donors. Thirteen studies included hospitalised and eleven convalescent patients. Only one study included children. IgG was assessed using eleven in-house and 16 commercial ELISAs. Neutralising antibodies were detected using the Focus Reduction Neutralisation Test (FRNT) by two studies, the PRNT by five studies, the Virus Neutralisation Test (VNT) by three studies, the surrogate VNT by one study, the pseudovirus VNT by nine studies and the microneutralisation method by seven studies.

The risk of bias in the studies is shown in the supporting information (S2 Table). The main risk identified was the lack of sample size estimations, as only two studies included pre-study sample size estimations, and sample sizes were usually based on the number of cases or samples available. None of the studies reported blinding of the endpoints. Four studies were considered to have fair and 23 to have good study quality.

### Proportion of participants with SARS-CoV-2 neutralising antibodies

Eighteen studies reported the proportion of participants with SARS-CoV-2 neutralising antibodies with titres >1:20 [9–26], 18 reported titres >1:40 [9–12,14,16,17,19–22,25,27–32],17 reported titres >1:80 [9–12,14,16,17,19–22,25,27,29,33–35], and 18 reported titres >1:160 [9–12,14–17,19–22,25,27,29,32–34]. The pooled proportion of participants with neutralising antibodies varied with the titre threshold used (Fig 2) and ranged from 85.3% (95%CI 83.5 to 86.9) in studies using threshold titres >1:20; 83.9% (95%CI 82.2 to 85.6) with titres >1:40, and 70.2% (95%CI 68.1 to 72.5) and 54.2% (52.0 to 56.5) with titres >1:80 and >1:160, respectively (Table 2).

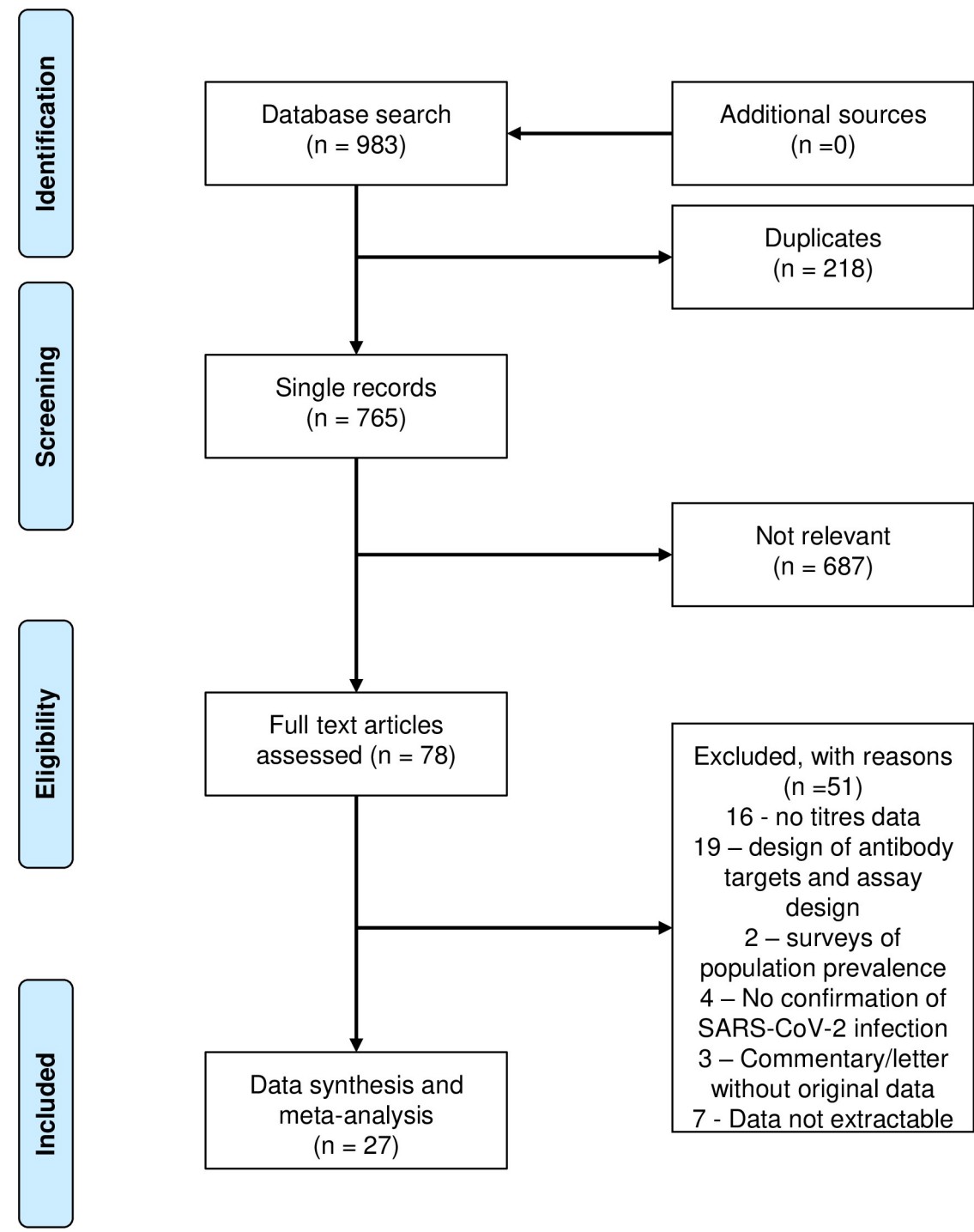

**Fig 1. Flow diagram of study selection.**

The proportion of participants classified as having neutralising antibodies varied with the detection method, with the pseudovirus method reporting the highest and the microneutralisation method the lowest proportion of participants detectable (titre threshold >1:20, 94.9% [95%CI 90.2 to 98.0] and 66.8% [95%CI 55.0 to 75.2], respectively), with similar differences

**Table 1. Study characteristics of the 27 studies included.**

| Author | Country | Study design | Setting | Participants | Disease Severity* | IgG Assay* | Target | Neutralising Antibody method | Sample size |
|---|---|---|---|---|---|---|---|---|---|
| Brochot et al | France | Cohort | H | Inpatient | M/S | ELISA IH | S1, S2, N, RBD | Pseudovirus VNT | 30 |
| Tang et al | USA | Cohort | H | Inpatient | M/S | Euroimmun Roche Abbott | S1 S N | FRNT | 48 |
| Kohmer et al | Germany | Case series | H | Inpatient | M/S | Euroimmun Vircell | S1 S, N | PRNT | 45 |
| Bonelli et al | Italy | Cohort | H | Inpatient/ Community | M/S | Diasorin | S1, S2 | Microneutralisation | 304 |
| Kohmer, Westhaus et al | Germany | Case series | H | Inpatient | S | Abbott Virotech Vircell | N N N | PRNT | 33 |
| Suhandynata et al | USA | Cohort | C | Convalescent | A | Abbott Roche Diazyme | N N, S n/a | Pseudovirus VNT | 63 |
| Zhang et al | China | Case series | H | Convalescent | M/S | ELISA IH1 IH2 | S1 S2 | Pseudovirus VNT | 67 |
| Stromer et al | Germany | Case series | PD | Convalescent | M/Md/S | Abbott Roche Dyazyme Mikrogen Epitope diagnostics Euroimmun | N N N, S N N S1 | PRNT | 26 |
| Liu et al | China | Case series | H | Paediatric | M/S | n/a | n/a | Pseudovirus VNT | 9 |
| Crawford et al | USA | Cohort | H/C | Convalescent | A/M/Md | ELISA IH1 IH2 | S | Psuedovirus VNT | 34 |
| Juno et al | Australia | Cohort | unstated | Convalescent | M | ELISA IH1 IH2 | S RBD | Microneutralisation | 41 |
| Wang et al | China | Cohort | H | Inpatient | M/S | ELISA IH1 IH2 IH3 IH4 IH5 | S1 S2 N RBD S | Pseudovirus VNT | 23 |
| Zeng et al | USA | Cohort | H | Inpatient | M/M | Epitope diagnostics | N | PseudovirusVNT | 55 |
| Ko et al | South Korea | Cohort | H | Inpatient | A | PCL | N, S | Microneutralisation | 15 |
| Ruetalo et al | Germany | Case series | PD | Community | A/M | Euroimmun Mediagnost Roche | S1 RBD N | VNT | 49 |
| Lee et al | USA | Cohort | PD | Convalescent | A/Md/Md/S | ELISA IH | RBD | PRNT | 149 |
| Wu et al | China | Case series | H | Inpatient | M | ELISA IH1 IH2 IH3 | RBD S1 S2 | Pseudovirus VNT | 175 |
| Bosnjak et al | Germany | Cohort | H | Inpatient | M | Euroimmun | S1 | Surrogate VNT | 40 |
| Jaaskelainen et al | Finland | Case series | PD | Convalescent | M/Md/S | Euroimm Diasorin Abbott | S1 S1, S2 n/a | Microneutralisation | 70 |
| Percivalle et al | Italy | Cohort | PD | Convalescent | A/M/Md/S | n/a | n/a | Microneutralisation | 38 |
| Suthar et al | USA | Case series | H | Inpatient | M/Md | ELISA IH | RBD | FRNT | 44 |
| Zettl et al | Germany | Cohort | H | Inpatient | M/Md/S | Euroimmun | S1 | VNT | 25 |

(*Continued*)

**Table 1.** (Continued)

| Author | Country | Study design | Setting | Participants | Disease Severity* | IgG Assay* | Target | Neutralising Antibody method | Sample size |
|---|---|---|---|---|---|---|---|---|---|
| Salazar et al | USA | Cohort | PD | Convalescent | M/Md/S | ELISA IH | S | Microneutralisation | 68 |
| Klein et al | USA | Case series | PD | Convalescent | M/Md | Euroimmun ELISA IH1 IH2 | S1 S RBD | Microneutralisation | 126 |
| Padoan et al | Italy | Cohort | H | Convalescent | M/Md/S | Abbott Roche DIESSE Ortho Clinical Dxs | N N Native antigen S | PRNT | 52 |
| Gozalbo et al | Spain | Case series | Hospital | Inpatient | Md/S | ELISA IH | RBD | Pseudovirus VNT | 51 |
| Mueller et al | Germany | Case series | Community | Community | M | Euroimmun Diasorin Abbott Roche | S1 S1, S2 N N | VNT | 34 |

*A = asymptomatic, M = mild, Md = Moderate, S = severe; IH = in house; hospital = H, Community = C, Plasma donor = PD; Bold = ELISA individual data extracted

reported at all titre cut-offs (Table 2). The pooled proportion positive varied with disease severity (Fig 3). Studies focusing on severe COVID-19 reported higher proportions with neutralising antibodies than studies focusing on mild COVID-19, especially if they had used titre thresholds >1:80 (84.8% [95%CI 80.0 to 89.2] and 56.7% [95%CI 49.9 to 62.9], for severe and mild COVID-19, respectively) or >1:160 (74.4% [95%CI 67.5 to 79.7] and 44.1% [95%CI 37.3 to 50.6], respectively). Similarly, the proportion of patients testing positive was lower in studies

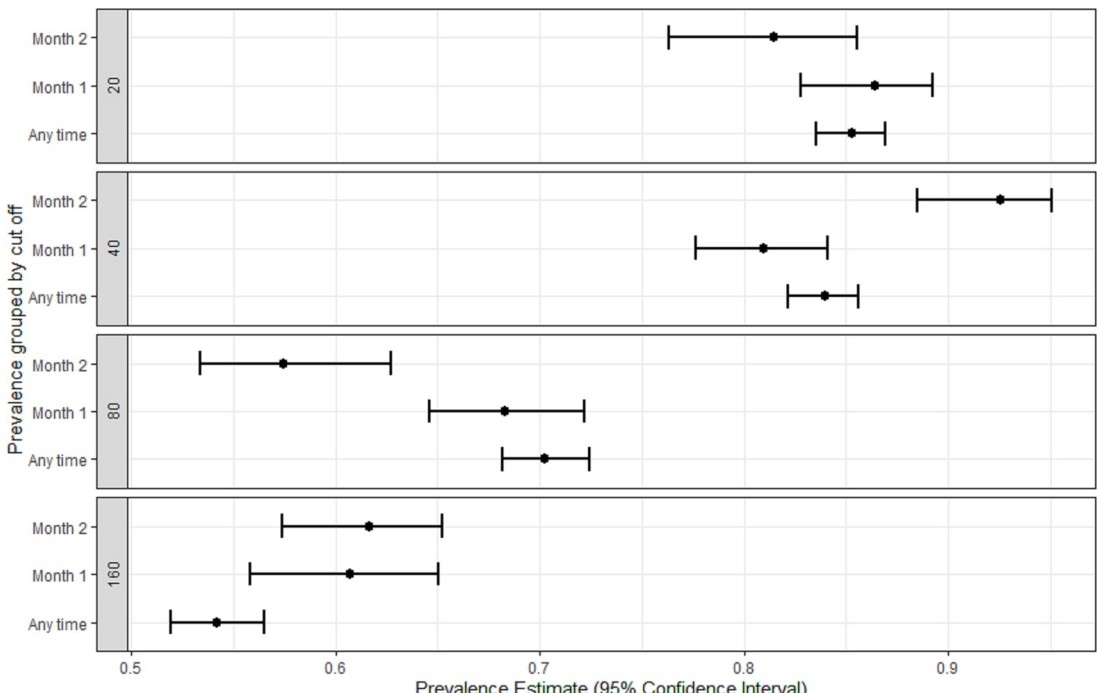

**Fig 2. Estimated pooled proportion (95% confidence interval) of participants with neutralisation antibodies by titre cut-off and time.**

**Table 2. Estimated proportion of participants with neutralising antibodies with titre cut off > 1:20, 1:40, 1:80 and 1:160 by timepoints and selected covariates.**

| | Cut off >20 | Cut off >40 | Cut off >80 | Cut off >160 |
|---|---|---|---|---|
| **Variables** | % (95% CI) | % (95% CI) | % (95% CI) | % (95% CI) |
| **Any time** | | | | |
| **Overall** | **85.3 (83.5–86.9)** | **83.9 (82.2–85.6)** | **70.2 (68.1–72.5)** | **54.2 (52.0–56.5)** |
| **Severity** | | | | |
| Asymptomatic | 14.4 (2.3–35.4) | 97.9 (89.1–100.0) | 28.6 (17.9–39.2) | 10.0 (3.7–20.1) |
| Mild | 81.2 (76.5–85.3) | 79.3 (74.2–84.2) | 56.7 (49.9–62.9) | 44.1 (37.3–50.6) |
| Mixed–excluding severe | 93.2 (85.1–97.9) | 87.1 (76.1–94.5) | 68.9 (59.6–77.9) | 53.1 (40.7–64.9) |
| Mixed–including severe | 92.8 (88.6–96.2) | 76.8 (72.9–80.2) | 88.0 (84.3–91.2) | 62.6 (57.7–68.5) |
| Moderate | 82.3 (70.5–90.7) | 86.1 (79.5–91.1) | 73.6 (62.1–82.4) | 57.6 (47.1–67.4) |
| Severe | 89.3 (83.7–93.6) | 90.0 (85.0–94.0) | 84.8 (80.0–89.2) | 74.4 (67.5–79.7) |
| Unknown | 76.4 (57.0–90.1) | – | – | – |
| **Setting** | | | | |
| Community | 63.4 (57.6–70.0) | 85.7 (80.6–89.9) | 53.5 (46.6–60.2) | 45.5 (39.9–52.4) |
| Plasma donor | 78.9 (75.3–82.1) | 66.8 (62.5–71.4) | 66.0 (61.8–70.1) | 41.1 (36.8–46.3) |
| Hospital | 93.4 (91.0–95.1) | 88.9 (86.7–90.9) | 75.1 (72.7–78.2) | 61.9 (58.9–64.8) |
| Hospital/Community | 93.6 (86.9–98.2) | – | – | 65.4 (57.8–89.6) |
| Not stated | – | 98.3 (91.9–100.0) | – | – |
| **Method** | | | | |
| VNT | 76.8 (72.4–80.6) | 81.3 (76.1–85.7) | 54.3 (49.3–58.7) | 43.4 (37.8–48.7) |
| Surrogate VNT | 96.0 (90.8–98.7) | – | – | – |
| Microneutralisation | 74.6 (69.5–79.1) | 72.3 (68.6–75.8) | 53.9 (49.4–58.3) | 31.5 (27.4–35.8) |
| FRNT | 95.1 (92.1–97.4) | 95.5 (91.5–98.1) | 88.4 (80.1–94.2) | 79.0 (72.3–84.8) |
| PRNT | 82.6 (77.2–87.6) | 86.2 (82.8–90.3) | 74.7 (69.6–82.9) | 59.1 (54.2–65.5) |
| Pseudovirus | 95.0 (91.0–97.5) | 95.0 (91.6–97.3) | 86.1 (82.7–89.0) | 43.4 (37.8–48.5) |
| **Month 1** | | | | |
| **Overall** | **86.4 (82.8–89.2)** | **80.9 (77.6–84.1)** | **68.3 (64.6–72.2)** | **60.7 (55.8–65.1)** |
| **Severity** | | | | |
| Asymptomatic | – | – | – | – |
| Mild | 77.3 (67.7–86.0) | 70.7 (60.7–80.3) | 54.1 (40.4–65.3) | 50.3 (38.4–61.6) |
| Mixed–excluding severe | 90.8 (77.7–98.0) | 87.5 (62.2–99.4) | 66.3 (35.1–90.7) | 65.5 (33.9–89.3) |
| Mixed–including severe | 93.7 (88.9–97.4) | 88.0 (79.1–95.0) | 72.6 (62.0–81.2) | 61.6 (53.9–74.2) |
| Moderate | 80.5 (70.7–90.2) | 74.1 (65.0–83.5) | 67.3 (58.2–77.2) | 58.1 (50.7–66.8) |
| Severe | 81.9 (65.6–90.2) | 79.0 (69.6–87.2) | 70.5 (62.1–79.1) | 65.0 (46.8–74.6) |
| Unknown | – | – | – | – |
| **Setting** | | | | |
| Community | 98.3 (92.9–100) | 98.8 (93.4–100.0) | 97.7 (90.2–100.0) | 97.6 (90.1–100.0) |
| Plasma donor | 66.9 (55.0–75.2) | 53.8 (45.2–61.8) | 33.9 (26.1–42.4) | 39.1 (29.7–46.6) |
| Hospital | 92.5 (89.1–95.1) | 90.0 (86.0–93.3) | 77.8 (72.5–82.4) | 70.6 (64.2–76.3) |
| Hospital/Community | 93.8 (87.8–97.9) | – | – | – |
| **Method** | | | | |
| VNT | – | – | 54.1 (40.2–69.4) | – |
| Surrogate VNT | – | – | – | – |
| Microneutralisation | 66.8 (55.0–75.2) | 53.8 (45.2–61.8) | 33.9 (26.1–42.4) | 32.7 (20.5–41.4) |
| FRNT | 94.0 (90.7–96.7) | 93.4 (88.5–96.8) | 81.0 (72.5–87.7) | 68.6 (59.8–76.2) |
| PRNT | 84.6 (75.3–92.2) | 84.6 (75.1–91.9) | 78.5 (68.5–87.3) | 65.6 (58.3–76.7) |
| Pseudovirus | 94.9 (90.2–98.0) | 92.0 (86.0–96.5) | 84.8 (77.5–90.6) | 80.4 (71.6–87.7) |
| **Month 2** | | | | |

(*Continued*)

**Table 2.** (Continued)

| Variables | Cut off >20 % (95% CI) | Cut off >40 % (95% CI) | Cut off >80 % (95% CI) | Cut off >160 % (95% CI) |
|---|---|---|---|---|
| **Overall** | **81.5 (76.3–85.6)** | **92.6 (88.5–95.1)** | **57.5 (53.4–62.7)** | **61.6 (57.4–65.2)** |
| **Severity** | | | | |
| Asymptomatic | – | – | 19.0 (4.4–41.7) | 6.9 (0.3–22.5) |
| Mild | 66.2 (53.5–78.5) | 87.7 (81.1 to 93.3) | 48.4 (39.1–57.9) | 25.8 (18.1–34.8) |
| Mixed–excluding severe | 97.3 (91.6–99.9) | 97.9 (92.3 to 99.9) | 91.8 (82.3–97.6) | 75.4 (62.8–85.8) |
| Mixed–including severe | 88.0 (70.6–95.4) | 88.7 (73.0 to 94.6) | 64.3 (60.3–68.1) | 77.7 (71.0–83.5) |
| Moderate | 74.0 (56.4–86.5) | 92.6 (77.8 to 99.6) | 85.6 (57.1–99.2) | 48.9 (17.0–79.5) |
| Severe | 75.3 (65.1–84.9) | 96.7 (90.0 to 99.7) | 75.4 (61.7–89.3) | 77.2 (64.6–88.3) |
| Unknown | – | – | – | – |
| **Setting** | | | | |
| Community | – | – | – | – |
| Plasma donor | 71.8 (64.1–78.8) | 87.2 (77.6–93.7) | 56.0 (49.2–60.8) | 70.4 (61.9–77.7) |
| Hospital | 87.3 (80.4–91.9) | 91.8 (85.7–94.8) | 57.9 (52.8–64.7) | 58.7 (53.6–62.9) |
| Hospital/Community | – | – | – | – |
| Not stated | – | 98.5 (93.2–100.0) | – | – |
| **Method** | | | | |
| VNT | – | – | 4.5 (0.6–12.1) | – |
| Surrogate VNT | – | – | – | – |
| Microneutralisation | 71.8 (64.1–78.8) | 93.7 (88.6–96.9) | 52.8 (47.3–57.9) | 37.3 (30.0–44.2) |
| FRNT | 77.3 (54.8–90.9) | 94.7 (75.9–100.0) | 40.0 (28.1–52.0) | 80.7 (60.3–93.0) |
| PRNT | 86.3 (78.1–92.6) | 73.3 (64.2–81.7) | 64.5 (59.4–69.6) | 71.6 (66.4–76.8) |
| Pseudovirus | 93.7 (90.0–96.9) | 97.5 (93.8–99.3) | 79.5 (70.0–95.3) | 73.9 (68.9–8.3) |

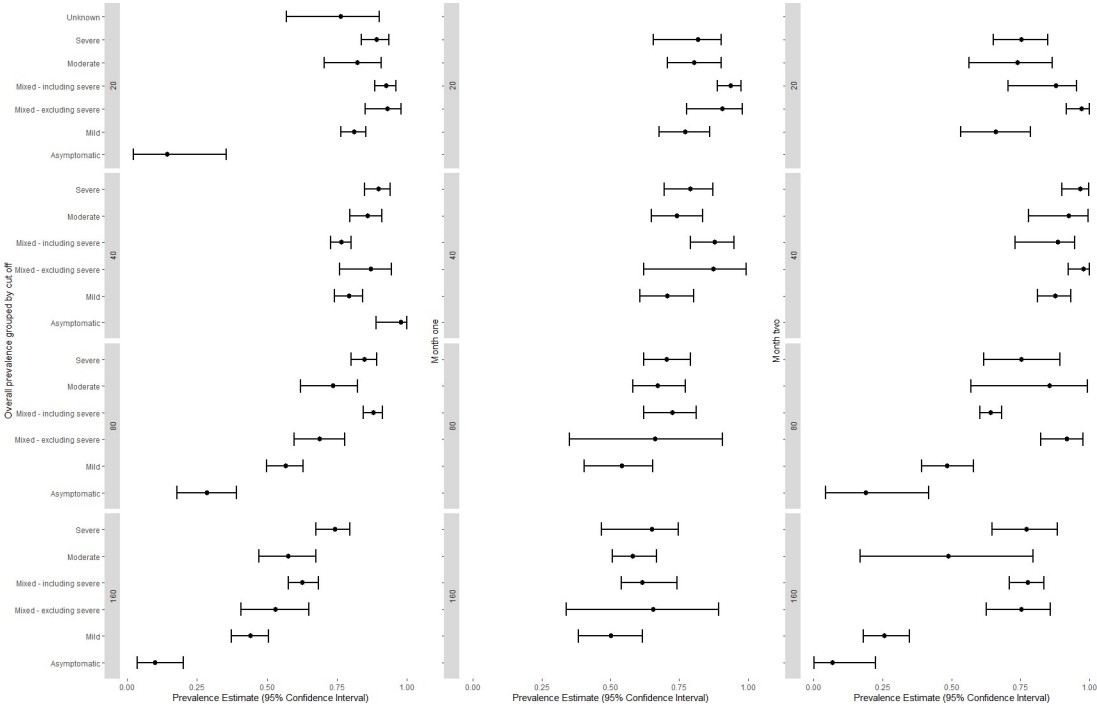

**Fig 3. Estimated proportion positive (95% confidence interval) by disease severity, titre cut-off and time.**

of asymptomatic infections (14.4% with a titre threshold >1:20 [95%CI 2.3 to 35.4]; 97.1% with a titre >1:40 [one study, 95%CI 89.1 to 100], 28.6% with >1:80 [95%CI 17.9 to 39.2] and 10.0% [3.7 to 20.1] with >1:160). These associations were reflected in studies recruiting patients in hospitals, which had higher proportions of participants with neutralising antibodies than those recruiting among plasma donors or the community (titres >1:20: 93.4% [95%CI 91.0 to 95.1] compared with 78.9% [95%CI 75.3 to 82.1] and 63.4% [95%CI 57.6 to 70.0], respectively).

Studies reporting the proportion of patients with neutralising antibodies one and two months after infection were analysed separately to assess the effect of time on the development of antibias during the early stages after infection. The pooled proportion with neutralising antibodies among plasma donors was higher in month two than in month one although this was not statistically significant for most thresholds (Table 2). In contrast, in studies recruiting from hospital settings, the proportion of participants with neutralising antibodies was higher in month two at titres > 1:80 (77.8% [95%CI 72.5 to 82.4] to 57.9% [95% CI 52.8 to 64.7], >1:160 70.6% [95%CI 64.2 to 76.3] to 58.7% [95%CI 53.6 to62.9]). The pooled prevalence one and two months after infection had the same overall pattern, with the proportions being higher among participants with severe than mild COVID-19 (Table 2). The proportion positive in month two was lowest with the microneutralisation method than with the FRNT, PRNT and pseudovirus methods at titres >1:160 (37.3% [95%CI 30.0 to 44.2]; 80.7% [95%CI 60.3 to 93.0], 71.6% [95%CI 66.4 to 76.8] and 73.9% [95%CI 68.9 to 78.3], respectively).

## Correlation of neutralising and IgG antibodies

The modelled correlation between the aggregate proportion of patients with measurable IgG against SARS-CoV-2 and neutralising antibodies was very low (0.055, corresponding to poor correlation). The estimate was not modified by the assays, titre cut-offs used or the study setting (Table 3). The correlation between individual IgG and neutralising antibodies ranged from 0.16 to 0.756 across the studies (Table 4 and Fig 4). The correlation between individual values was higher than the correlation of aggregated data, but there was a high variability across studies, as shown in Fig 5. We were unable to explore whether the IgG ELISA or neutralising method used modified the correlations as each study used a unique set of IgG and

**Table 3. Maximum likelihood estimates (95% confidence interval) of the bivariate binomial mixed model.** The estimates for the intercept, titre cut off, setting and method are reported on the log-odds scale.

| Covariate | Immunoglobulin G | Neutralizing antibodies |
|---|---|---|
| Intercept | 5.398 (0.502 to 7.726) | 2.903 (0.206 to 5.112) |
| Titre cut off | 0.003 (-0.002 to 0.029) | -0.003 (-0.008 to 0.003) |
| Setting (ref. "Community") | | |
| Hospital | -1.640 (-4.051 to 0.122) | -0.298 (-1.874 to 1.305) |
| Not stated | 18.936 (0.334 to 56.965) | 1.401 (-0.957 to 4.611) |
| Plasma donor | -1.658 (-3.996 to 0.137) | -1.163 (-2.994 to 0.574) |
| Method for nAB (ref. "FRNT") | | |
| Microneutralisation | -0.504 (-2.220 to 1.230) | |
| PRNT | -0.782 (-2.436 to 0.757) | |
| Pseudovirus | 0.639 (-2.176 to 0.820) | |
| VNT | -0.472 (-2.256 to 1.373) | |
| Random effects | | |
| Variance | 3.762 (0.217 to 5.001) | |
| Correlation | 0.055 ($4.793 \times 10^{-9}$ to 0.500) | |

**Table 4. Maximum likelihood estimates of the correlation of IgG and neutralising antibody titres (nAB).** Estimates reported in a log-odds scale with 95% confidence intervals.

| Study | Correlation Estimate (95% Confidence intervals) |
|---|---|
| Ruetalo et al | 0.727 (0.561 to 0.848) |
| Kohmor et al | 0.589 (0.383 to 0.768) |
| Zettl et al | 0.16 (0.01 to 0.776) |
| Severity (Baseline = Mixed without severe) | IgG: Severe 1.094 (0.491 to 1.697) |
| | nAB: Severe 2.363 (1.66 to 3.066) |
| Suthar et al | 0.756 (0.589 to 0.87) |
| Jaaskaleinen et al | 0.375 (0.155 to 0.662) |
| Severity (Baseline = Mild) | IgG: Moderate -0.669 (-1.55 to 0.211) |
| | IgG: Severe -0.608 (-1.545 to 0.33) |
| | nAB: Moderate -0.897 (-1.772 to -0.022) |
| | nABs: Severe -0.693 (-1.625 to 0.239) |
| Zhang et al | 0.652 (0.503 to 0.776) |
| Severity (Baseline = Mixed without severe) | IgG: Severe 0·973 (0.321 to 1.626)<br>nAB: Severe 1.141 (0.611 to 1.67) |
| Salazar et al | 0.51 (0.356 to 0.661) |
| Severity (Baseline = Mild) | IgG: Severe 1.549 (0.919 to 2.179) |
| | nAB: Severe 1.401 (0.676 to 2.127) |
| Mueller et al | 0.723 (0.53 to 0.858) |
| Severity (Baseline = Asymptomatic) | IgG: Mild 1.023 (0.275 to 1.771) |
| | nAB: Mild 2.543 (1.232 to 3.853) |
| Wang et al | 0.597 (0.431 to 0.743) |
| Severity (Baseline = Mild) | IgG: Severe 0.564 (0.12 to 1.008) |
| | nAB: Severe 1.067 (0.614 to 1.521) |

IgG: immunoglobulin G. nABs: neutralising antibodies.

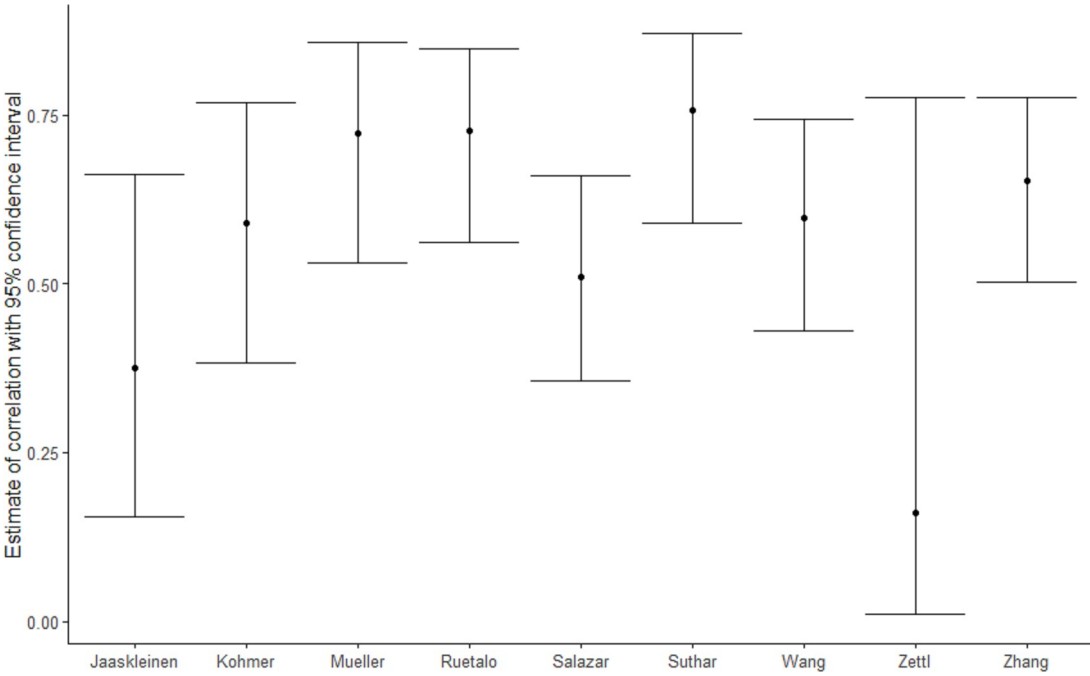

**Fig 4. Estimated correlation (95% confidence interval) of individual IgG and neutralising antibodies by study.**

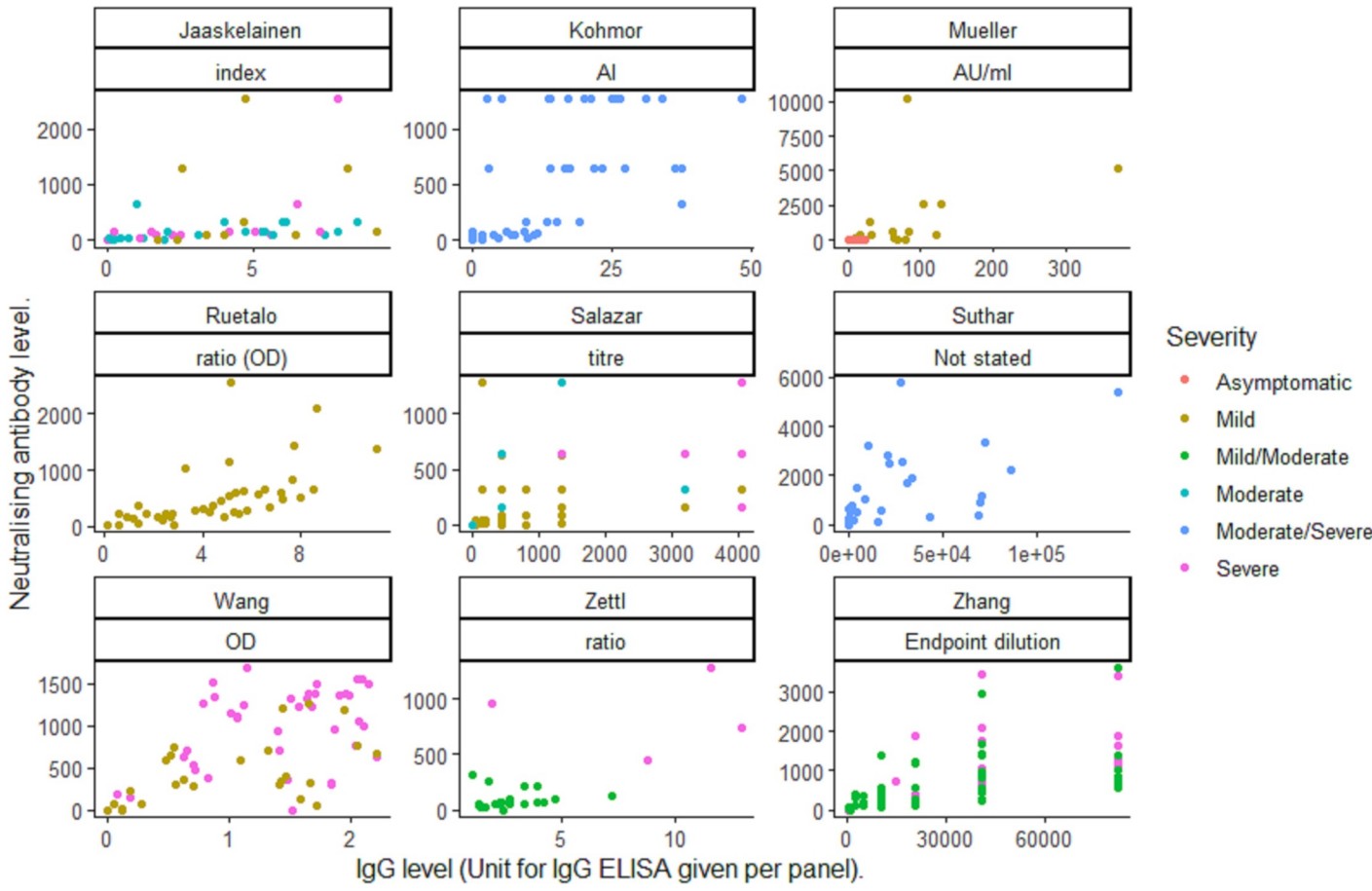

**Fig 5. Scatterplot of individual neutralising and IgG antibody values by study.**

neutralisation methods. In five of six studies reporting disease severity [9–11,19,21,25], there was an increased correlation between IgG and neutralising antibodies with increasing disease severity [10,11,19,21,25].

## Discussion

Despite major efforts to understand the mechanisms for immunity after COVID-19, this is the first comprehensive systematic review synthesising the proportion of individuals who exhibit neutralising antibodies after natural SARS-CoV-2 infection. Although 85% of participants with previous SARS-CoV-2 infection had detectable neutralising antibodies, there was a wide variation across studies, which was partly explained by the method and cut off titres used. Studies using the microneutralisation and the pseudovirus methods reported the lowest and highest proportion of participants with neutralising antibodies, respectively, while studies using low titre cut offs (i.e., >1:20 and >1:40) reported a higher proportion of participants as having neutralising antibodies than those using higher titre cut offs (i.e., >1:80 and >1:160).

The proportion of participants with neutralising antibodies varied with study setting, COVID-19 severity, and time since infection. Studies on severe COVID-19 reported higher proportions of patients with neutralising antibodies than studies focusing on mild and moderate COVID-19, with studies on asymptomatic infections having the lowest proportion of patients with antibodies. Similarly, hospital-based studies reported higher proportions of

participants with neutralising antibodies than studies based in the community or plasma donors, suggesting the study setting is a surrogate marker for disease severity. The lack of neutralising antibodies in a small but significant group of patients seen in this review might be explained by some patients having responses confined to SARS-CoV-2 antigens not identified by the assays used, or mediated through T cells, which are not detected by the neutralisation assays in this meta-analysis. Mild infections may also elicit responses that are restricted to the mucosal cells, where defence responses are dominated by the secretory immune system [36].

Only a subset of studies reported the pooled proportion of neutralising antibodies one and two months after infection. The proportion of participants with neutralising antibodies in the second month was slightly higher among plasma donors and slightly lower among patients recruited from hospital. However, the differences were small and confidence intervals overlapped. As neutralising antibodies peak within 21 days after symptoms onset, and symptoms onset if often poorly documented, it may be that the one-month timepoint is not ideal to demonstrate differences over time [37].

Surprisingly, the correlation between the detection of IgG and neutralising antibodies at study level was low, with a better correlation for the subset of studies with individual data. This is not surprising, as the correlation of aggregate data is necessarily a less sensitive analysis than individual data that can show higher definition. The correlation of individual IgG levels increased with disease severity, reaching an $r$ value of 0.756, but with a high variation between studies. This lack of consistency can be attributed to the neutralisation and ELISA assays used. Neutralisation assay sensitivity varies across the methods and ELISAs are based on different SARS-CoV-2 antigens (*e.g.* nucleoprotein or spike protein domains), which affect performance [38].

Despite neutralising antibodies not being detectable in a significant proportion of participants, these antibodies are only a visible fraction of the defence mechanism against COVID-19 and second infections are rare and adaptive immunity mechanisms involving B and T cells and mucosal neutralising antibodies are at play [39]. For example, memory B cells display clonal turnover after six months, with maturation of response and expressed antibodies having greater somatic hypermutation, increased potency, and resistance to the Spike protein receptor-binding domain mutations, indicative of a continued evolution of the humoral response [39].

We have examined variations resulting from study settings, testing methodologies, and disease severity and provide an overview based on a large number of studies, resourcing to individual data whenever possible. We have shown that current information is based on a multitude of methods assessing neutralising antibodies, a large variety of ELISAs, including in-house methods without WHO endorsement, and that these differences lead to different proportions of individuals being classified as having positive responses and, possibly, the poor correlation between IgG and neutralising antibodies in aggregated datasets. These limitations highlight the need for standardisation of the methodology and the development of guidelines for future studies. There is a similar a lack of standardisation of objective markers on the assessment of disease severity (e.g., C Reactive Protein) and results are often presented without stratification, which made it difficult to perform meta-regression of subgroups. Moreover, data on circulating SARS-CoV-2 variants, patients' age, outcome, T cell responses, ELISA's spike glycoprotein targets, and other potential covariates were not available in significant numbers and there were no studies from South America, the Indian subcontinent, or Africa, and results cannot be generalised to these populations.

In the event of a new pandemic virus there is a limited window where the natural epidemiology of the virus can be observed prior to treatment or vaccination. Without large cohort studies only, a fragmented picture was available, and ongoing cohorts will need to account for

virus variants, treatment and vaccination. Combining studies and performing a meta-analysis allowed an analysis of data from early stages and gives a more complete picture of the natural immunity at the beginning of the pandemic.

In conclusion, a high proportion of individuals have evidence of neutralising antibodies after SARS-CoV-2 infection. These proportions vary with disease severity, with asymptomatic infections being less likely to have detectable antibodies than those experiencing severe COVID-19. Most diagnostic tests, therapeutics, and vaccines are aimed at the SARS-CoV-2 Spike, and virus evasion occur through mutations that escape neutralising antibodies. Thus, research is needed to establish whether the lack of detectable neutralising antibodies interrelates with virus escapees with or without a vaccine [40]. This is particularly relevant in light of the recent dominance in parts of the UK of a strain with mutations in the spike protein that is associated with increased transmissibility [41]. Moreover, the minimum level of neutralising antibodies required to achieve protection is unknown at this stage, and immunological memory mechanisms may rapidly boost generation of antibodies that are not detectable in the peripheral blood in the absence of stimulation. This review also highlights the need for guidance on standardised protocols for the measurement of neutralising antibodies; for longitudinal studies to document how neutralising antibodies and their relationship with IgG levels change over time, and whether minimal or undetectable levels indicate lack of clinical protection and thus vulnerability to infection.

## Supporting information

**S1 Table. Full search strategy.**
(DOCX)

**S2 Table. Risk of bias of the studies included using the Quality Assessment Tool for Cohort and Case Series Studies of the National Institutes of Health (NIH).**
(DOCX)

**S1 Text. Statistical analysis.**
(DOCX)

## Author Contributions

**Conceptualization:** Helen R. Savage, Victor S. Santos, Thomas Edwards, Sanjeev Krishna, Timothy D. Planche, Joseph R.A. Fitchett, Daniela E. Kirwan, Emily R. Adams, Luis E. Cuevas.

**Formal analysis:** Helen R. Savage, Victor S. Santos, Thomas Edwards, Emanuele Giorgi, Henry M. Staines, Luis E. Cuevas.

**Funding acquisition:** Sanjeev Krishna, Henry M. Staines, Joseph R.A. Fitchett, Luis E. Cuevas.

**Investigation:** Helen R. Savage, Victor S. Santos, Thomas Edwards.

**Methodology:** Helen R. Savage, Victor S. Santos, Thomas Edwards, Emanuele Giorgi, Ana I. Cubas Atienzar, Emily R. Adams, Luis E. Cuevas.

**Project administration:** Luis E. Cuevas.

**Supervision:** Daniela E. Kirwan, David J. Clark, Emily R. Adams, Luis E. Cuevas.

**Writing – original draft:** Helen R. Savage, Victor S. Santos, Thomas Edwards, Emanuele Giorgi, Ana I. Cubas Atienzar, David J. Clark, Luis E. Cuevas.

**Writing – review & editing:** Helen R. Savage, Victor S. Santos, Thomas Edwards, Emanuele Giorgi, Sanjeev Krishna, Timothy D. Planche, Henry M. Staines, Joseph R.A. Fitchett, Daniela E. Kirwan, Ana I. Cubas Atienzar, David J. Clark, Emily R. Adams, Luis E. Cuevas.

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
