## [Decision Letter · Decision Letter 0]

23 Apr 2021

Dear Professor Cuevas,

Thank you very much for submitting your manuscript "Prevalence of neutralising antibodies against SARS-CoV-2 in acute infection and convalescence: a systematic review and meta-analysis" for consideration at PLOS Neglected Tropical Diseases. As with all papers reviewed by the journal, your manuscript was reviewed by members of the editorial board and by several independent reviewers. In light of the reviews (below this email), we would like to invite the resubmission of a revised version that takes into account the reviewers' comments. 

We cannot make any decision about publication until we have seen the revised manuscript and your response to the reviewers' comments. Your revised manuscript is also likely to be sent to reviewers for further evaluation.

Sincerely,

Tao Lin, DVM, MSc

Associate Editor

Liesl Zuhlke

Deputy Editor

Reviewer's Responses to Questions

**Key Review Criteria Required for Acceptance?**

**Methods**

-Are the objectives of the study clearly articulated with a clear testable hypothesis stated?

-Is the study design appropriate to address the stated objectives?

-Is the population clearly described and appropriate for the hypothesis being tested?

-Is the sample size sufficient to ensure adequate power to address the hypothesis being tested?

-Were correct statistical analysis used to support conclusions?

-Are there concerns about ethical or regulatory requirements being met?

Reviewer #1: The study is clearly articulated and the objectives well described. The design is appropriately set for a study that is needed in the field. The studies selected are often small and a larger size would benefit however the analysis provides clear conclusions using well established methodology.

**Results**

-Does the analysis presented match the analysis plan?

-Are the results clearly and completely presented?

-Are the figures (Tables, Images) of sufficient quality for clarity?

Reviewer #1: The analysis presented match the analysis plan. The neutralizing antibody results are clearly and well presented.

The IgG analysis comes with a lot of complexity is lacking granularity.

Reviewer #2: (No Response)

**Conclusions**

-Are the conclusions supported by the data presented?

-Are the limitations of analysis clearly described?

-Do the authors discuss how these data can be helpful to advance our understanding of the topic under study?

-Is public health relevance addressed?

Reviewer #1: The conclusions are supported great deal by the data presented and the limitations of analysis are described. The authors discuss how these data are helpful to advance our understanding of the neutralizing antibody immunity against SARS-CoV2. However, larger studies are needed to be able to address the relevance for public health recomendations.

Reviewer #2: (No Response)

**Editorial and Data Presentation Modifications?**

Reviewer #1: Results

THe IgG/nAntibodie association analysis needs differentiation between different viral antigen or limit it to the surface antigen against which the neutralizing antibodies are directed against.

Discussion

“The lack of neutralising antibodies in a small but significant group of patients might be explained by some patients having responses confined to other SARS-CoV-2 antigens or mediated through T cells, which are not detected by the neutralisation assays in this meta-analysis.” It is not clear who those groups of patients are.

“Mild infections may also elicit responses that are restricted to the mucosal cells, where defence responses are dominated by the secretory immune system.” Interesting statement and good hypothesis that is not evidenced.

“The proportion of participants with neutralising antibodies in the second month was slightly higher among plasma donors and slightly lower among patients recruited from hospital. However, the differences were small and confidence intervals overlapped.

Moreover, neutralising antibodies peak within 21 days after symptoms onset and the one month is not ideal to demonstrate time differences” Do you mean that antibodies in hospitalized individuals might decline faster? Do the responses peak within 21 days for both groups?

Reviewer #2: none

**Summary and General Comments**

Reviewer #1: The authors describe a meta analysis of published studies describing the neutralising antibody titres induced by SARS-CoV 2 acute infection. This is an important study considering the need to understand corelates of immunity by both infection and vaccination. Furthermore, increasing evidence is arguing for the role of neutralizing antibodies. One element that is clearly understood throughout the study is the lack of large cohort comprehensive studies to establish this fact. Only very few studies could be retained for the analysis amongst large number of publications initially selected. None the less the analysis is well conducted, and the authors conclude that the disease severity is largely correlating with the levels of neutralizing antibodies induced. 

There are differences between neutralization assay sensitivity that also show throughout the study and it is unclear to what degree it will affect the analysis.

The authors have investigated the associations between neutralizing antibody titres and the Elisa IgG titres. This part of the study has some weakens as besides the neutralising antibody assay variation the binding antibody assays can also be very variable. The type of assay used and most of all the origin of recombinant antigen utilised will influence the assay a great deal. In addition, table 1 indicates a variety of viral antigens utilised to measure IgG and the study lacks granularity.

Reviewer #2: (No Response)

PLOS authors have the option to publish the peer review history of their article (what does this mean?). If published, this will include your full peer review and any attached files.

Reviewer #1: No

Reviewer #2: No
---

## [Decision Letter · Decision Letter 1]

9 Jun 2021

Dear Professor Cuevas,

We are pleased to inform you that your manuscript 'Prevalence of neutralising antibodies against SARS-CoV-2 in acute infection and convalescence: a systematic review and meta-analysis' has been provisionally accepted for publication in PLOS Neglected Tropical Diseases.

Best regards,

Tao Lin, DVM, MSc

Associate Editor

Liesl Zuhlke

Deputy Editor

Reviewer's Responses to Questions

**Key Review Criteria Required for Acceptance?**

**Methods**

-Are the objectives of the study clearly articulated with a clear testable hypothesis stated?

-Is the study design appropriate to address the stated objectives?

-Is the population clearly described and appropriate for the hypothesis being tested?

-Is the sample size sufficient to ensure adequate power to address the hypothesis being tested?

-Were correct statistical analysis used to support conclusions?

-Are there concerns about ethical or regulatory requirements being met?

Reviewer #2: Yes

**Results**

-Does the analysis presented match the analysis plan?

-Are the results clearly and completely presented?

-Are the figures (Tables, Images) of sufficient quality for clarity?

Reviewer #2: Yes

**Conclusions**

-Are the conclusions supported by the data presented?

-Are the limitations of analysis clearly described?

-Do the authors discuss how these data can be helpful to advance our understanding of the topic under study?

-Is public health relevance addressed?

Reviewer #2: Yes

**Editorial and Data Presentation Modifications?**

Reviewer #2: (No Response)

**Summary and General Comments**

Reviewer #2: (No Response)

PLOS authors have the option to publish the peer review history of their article (what does this mean?). If published, this will include your full peer review and any attached files.

Reviewer #2: No

---

## [Editor Report · Acceptance letter]

2 Jul 2021

Dear Professor Cuevas,

We are delighted to inform you that your manuscript, "Prevalence of neutralising antibodies against SARS-CoV-2 in acute infection and convalescence: a systematic review and meta-analysis," has been formally accepted for publication in PLOS Neglected Tropical Diseases.

Best regards,

Shaden Kamhawi

co-Editor-in-Chief

Paul Brindley

co-Editor-in-Chief
